# Evaluation of Phytotoxic and Cytotoxic Effects of Prenylated Phenol Derivatives on Tomato Plants (*Solanum lycopersicum* L.) and *Botrytis cinerea* B-05 Spores

**DOI:** 10.3390/plants14213277

**Published:** 2025-10-27

**Authors:** Gerard Núñez, Ligia Llovera, Dioni Arrieche, Romanet Berrios, Mauricio Soto, Mauricio Osorio-Olivares, Andrés F. Olea, Efraín Sarmiento, Azucena González, Héctor Carrasco, Lautaro Taborga

**Affiliations:** 1Departamento de Química, Universidad Técnica Federico Santa María, Valparaíso 2340000, Chile; gerard.nunez@sansano.usm.cl (G.N.); ligia.llovera@usm.cl (L.L.); dioni.arrieche@sansano.usm.cl (D.A.); romanet.berrios@usm.cl (R.B.); mauricio.sotoc@usm.cl (M.S.); 2Grupo QBAB, Instituto de Ciencias Aplicadas, Facultad de Ingeniería, Universidad Autónoma de Chile, Av. del Valle Sur 534, Santiago 8580640, Chile; osomauricio@gmail.com (M.O.-O.); andres.olea@uautonoma.cl (A.F.O.); efrain.sarmiento@cloud.uautonoma.cl (E.S.); 3Instituto de Ciencias Agrarias, Consejo Superior de Investigaciones Científicas, 28006 Madrid, Spain; azu@ica.csic.es

**Keywords:** *B. cinerea*, mycelial growth, cytotoxicity, allylphenols, prenylphenols, seed germination, root growth

## Abstract

The phytopathogenic fungus *Botrytis cinerea*, which causes gray mold disease, has become a limiting factor on agricultural production. *B. cinerea* field control is made mainly using chemical fungicides, which has led to the spreading of resistant populations of this fungus. Thus, the quest of new fungicides molecules has been focused on synthesis of natural product-inspired compounds. The main aim of this work is to synthesize prenylated phenol derivatives and to assess their potential application as antifungal agents with minimal phytotoxic effects. Thus, new prenylphenols (**4**, **5**, and **7**) have been obtained by microwave irradiation with yields ranging from 2.4% to 42.9%, whereas compounds **8** and **9** were synthesized with yields of 25.6% and 54.1%, respectively. The effect of different concentrations of these compounds on *B. cinerea* spore germination, and their phytotoxic effect on tomato (*Solanum lycopersicum* L.) seed germination and root growth, were evaluated. Obtained results indicate that biological activities of all tested compounds are concentration-dependent. Interestingly, compound **7** exhibits the highest antifungal activity against *B. cinerea* spores (IC_50_ < 50 µg/mL) with minimal phytotoxicity on tomato seed germination and root growth. In contrast, compounds **2** and **3** are active against spores (IC_50_ = 461 and 325 µg/mL, respectively) but, at the same time, their phytotoxicity is important at the highest concentrations. These results indicated that the presence of hydroxyl and methyl substituents on the aromatic ring of these compounds induces variations in biological activities, and compound **7** could be a promising candidate as a sporicidal agent.

## 1. Introduction

The control of *Botrytis cinerea*, a phytopathogenic fungus responsible for the so-called gray mold disease, is of great economical relevance for agricultural industry because this infection can affect crops such as grapes, strawberries, raspberries, lettuce, cucumbers, beans, tomatoes, flowers, and forest nurseries [1,2], in a wide range of conditions and geographic areas, either in open fields or in greenhouses [3]. It is also capable of ruining agricultural products after harvest, as it can be active at very low temperatures [4,5].

Currently, *B. cinerea* field control is made mainly using chemical fungicides, which has led to the appearance of resistant populations of this fungus [6,7,8]. To overcome this problem, during the last decades, much work has been dedicated to developing new fungicides molecules based on active natural products or their synthetic derivatives [9,10,11,12,13,14]. This alternative approach for the control of phytopathogenic fungus is favored by their low toxicity as well as minimal environmental impact. Thus, the quest of new fungicides molecules has been focused on the synthesis of natural product-inspired compounds. Among these, allyl, prenyl, and geranylphenols are especially attractive due to their diverse biological activities, including antioxidants [15,16,17,18], anti-inflammatory [19,20,21], antineoplastic [22,23], antibacterial [24,25,26,27], and antifungal activities [28].

In previous work, phenol, allylphenol, and geranylated phenol derivatives with various substituents on the aromatic ring have been synthesized and tested for antifungal activity. Results indicate that eugenol inhibits mycelial growth of a virulent, multidrug-resistant strain of *B. cinerea* (PN2), with IC_50_ values ranging from 31 to 95 ppm [11]. On the other hand, it has been shown that phenol derivatives with allyl or geranyl chains attached in *ortho* position to the hydroxyl group exhibit biological activity against phytopathogenic fungi such as *B. cinerea* [12,29,30,31,32] and *Phytophthora cinnamomic* [12].

Structure–activity analysis suggests that antifungal activity on *B. cinerea* might be related with the lipophilic character of these compounds, which determines their accumulation in the fungal membrane. Therefore, the aim of this study was to synthesize a series of prenylated phenols and their hydrated derivatives, and to evaluate their antifungal activity against *B. cinerea* spores and phytotoxicity on tomato (*Solanum lycopersicum* L.) seeds and roots, to identify structural features that enhance antifungal efficacy while minimizing phytotoxic effects. All synthesized compounds were tested for inhibition of *B. cinerea* spores and phytotoxicity on tomato fruits and roots.

## 2. Results and Discussion

A series of prenylated phenol derivatives and two hydrated prenylated phenols have been synthesized to evaluate their inhibitory effect on spore germination of *B. cinerea*. These compounds are shown in Figure 1.

Prenyl and geranylphenols are, commonly, synthesized by slow addition of prenol or geraniol to a solution of phenols in anhydrous media and in the presence of BF_3_·OEt_2_ as a catalyst [16,29,33]. For example, using this catalyst and diethyl ether/CH_2_Cl_2_ as solvent, compounds **1**–**3** and **6** have been previously synthesized with yields ranging from 15% to 29% [16]. It has also been shown that by carrying out this reaction under microwave irradiation, reaction times for coupling of geraniol and phenols decrease whereas yields increase [34].

### 2.1. Synthesis

Herein, prenylated phenols **1**–**7** were obtained by direct coupling of prenol (3-methyl-2-buten-1-ol) and the corresponding phenol derivatives using ZnCl_2_ as catalyst, ethyl acetate (AcOEt) as solvent, and microwave irradiation at 60 °C. Figure 1 illustrates the coupling reaction between 2-methylresorcinol and prenol, leading to compound **5**.

Following this method, new prenylated phenols, **4**, **5**, and **7**, were synthesized with yields 2.4%, 42.9%, and 17.4%, respectively, whereas compounds **1**–**3** and **6** were obtained with yields ranging from 4.4 to 13.7%. This result is interesting because it was expected that microwave irradiation would increase the yields while it decreases the time reaction. But herein, only monoprenylated phenols were obtained with lowest yields, whereas it has been reported that, at room temperature, diprenylated phenols and chromans are also obtained. For example, under the reaction conditions given in Figure 1, compound **2** is obtained as the only product with 18.8% yield. The same reaction at room temperature leads to **2**, diprenylated phenol and chroman, with 23%, 11%, and 15% yields, respectively [16]. In summary, the yields of this coupling reaction are higher at room temperature even though other products are formed. In the microwave-promoted reaction, only one product is formed but the lowest yields are obtained. However, the high reaction yield obtained on synthesis of **5** suggests that modification of reaction conditions used in this work could be a real improvement on the efficiency of Friedel–Craft coupling reaction.

The synthesis of hydrated prenylated phenols was performed by following a modified method previously reported for hydration of double bonds on the geranyl chain of several geranylated phenols [29]. As depicted in Figure 2, the hydration of prenyl chain was carried out using *p*-toluenesulfonic acid as catalyst in PEG-400/H_2_O and heating to 100 °C for 30 min, instead of PTSA-PB in dioxane/H_2_O min [34].

The use of PEG 400 as solvent is justified by its ability to stabilize the intermediate carbocation and therefore enhancing the nucleophilic water addition to the double bond. However, only compounds **3** and **7** were successfully hydrated, giving hydrated prenylated phenols **8** and **9** with yields of 26% and 54%, respectively. The successful hydration of these compounds indicates that the position of the prenyl chain plays a crucial role in the hydration reaction. Probably, the highest reactivity of **3** and **7** is due to the electron-donating effect of the hydroxyl group in *para* position in respect to the prenyl group. This could increase the electron density on the double bond, making it more reactive towards a potential electrophile and thereby facilitating the hydration of the double bond. On the other hand, the higher yield of compound **9** in respect to **8** could be attributed to the steric effect exerted by the methyl group neighboring the prenyl chain, which could be restraining the reaction on the double bond.

#### Structural Determination of Prenylated 2-Methylresorcinol (**5**)

Chemical structures of all synthesized compounds were primarily determined by ^1^H and ^13^C-NMR spectroscopic analysis supplemented by 2D NMR spectroscopy. Thus, all types of H and C atoms were identified and correlations between them were established.

For compound **5**, aromatic signals of H-5 and H-6 atoms were observed as two doublets at δ_H_ = 6.80 (dd, *J* = 8.8; 0.6 Hz, 1H) and δ_H_ = 6.34 (d, *J* = 8.4 Hz, 1H); besides, the coupling constant’s value indicates that both H are in *ortho* position. These results confirm mono substitution in the aromatic ring (Appendix A, Appendix A shows 1D-NMR spectrum of **5** using CDCl_3_ as deuterated solvent).

Furthermore, in the HMBC spectrum, the signal at δ_H_ = 3.30 ppm assigned as H-1′ (d, *J* = 7.2 Hz, 2H) shows interactions at ^2^*J*_H-C_ with C-2′ and C-4 (δ_C_ = 122.2 ppm and 118.4, respectively), as well as interactions at *^3^J*_H-C_ to C-3, C-3′, and C-5 at δ_C_ = 153.2 ppm, 135.0, and 126.8, respectively. These correlations in the HMBC spectra provided critical information on the connectivity of the prenyl chain in the aromatic ring, with coupling at C-4 being confirmed (Appendix A, Appendix A shows main heteronuclear correlations 2D HMBC at ^2^*J*_CH_ (red) and ^3^*J*_CH_ (blue) observed for compound **5**).

Finally, selective 1D NOESY NMR experiments show correlations between H-1′ and H-6, H-2′, and H-5′ (Appendix A, Appendix A shows selective 1D *NOESY* correlation obtained for compound **5**), which indicate that these protons are closely located within the molecule. Using this information, a three-dimensional structure was established for molecule **5**. In addition, using this technique, it was possible to find a *cisoid* relationship of the prenyl chain in respect of the methyl group (H-5′).

### 2.2. Bioactivity Assays

#### 2.2.1. Cytotoxicity Assay on *B. cinerea* Spores

In this study, antifungal activity of prenylated phenols, with different substituent patterns on aromatic rings, was evaluated by measuring their inhibitory effects on *B. cinerea* spore germination. Cytotoxicity activity on *B. cinerea* B-05 spores was determined using the MTT bioassay [35] in the absence and presence of compounds **1**–**9**. Typical results obtained for compound **5** are shown in Appendix A of the Appendix A. This figure specifically presents the cytotoxicity assay of compound **5** on *B. cinerea* (B-05) spores.

Measurements of spore germination as a function of tested concentrations, i.e., 800, 400, 200, 200, 100, and 50 μg/mL, can be used to plot dose–response curves from which IC_50_ values can be obtained. Results obtained for all tested compounds are shown in Table 1.

The data indicate that the position of both hydroxyl and methyl groups affects, significatively, the antifungal activity of these compounds.

Prenylated phenols studied herein have two hydroxyl groups attached to the aromatic ring, and the data in Table 1 suggest that the relative position of hydroxyl groups play a very important role on activity. The highest activity is shown by compounds **3** and **7** with hydroxyl groups in a relative *ortho* position (catechol derivatives), whereas compounds **1** and **4** where hydroxyl groups are in *para* position (hydroquinone derivatives) have no activity. Compounds **2**, **5,** and **6**, with hydroxyl groups in *meta* position (resorcinol derivatives) exhibit intermediate activity. Different distributions of hydroxyl groups in benzene induce big changes in the dipole moments of catechol, resorcinol, and hydroquinone, i.e., it decreases as the hydroxyl groups move further apart. Thus, the dipole moments and, consequently, the polarity of these molecules, follow the order *para* < *meta* < *ortho* [36]. Then, attachment of a non-polar alkenyl chain to the aromatic ring in prenyl catechol derivatives would create two regions with distinct polarity within the molecule. This lipophilic character could enhance their interaction with the hydrophobic core of the fungal membrane, which might also facilitate the entry of these compounds into fungal cells. Their accumulation in the fungal membrane could interfere with cellular processes or induce structural changes which led to the release of reactive oxygen species. This mechanism of action has been proposed to explain antifungal activity of eugenol and allyl derivatives [11,37]. In addition, comparison of activities observed for **2** and **5**, **2** and **6**, and **3** and **7** indicate that incorporation of a methyl group in *ortho* position relative to the prenyl induces an increase in activity, i.e., the IC_50_ values of **2** and **5** are similar (460 μg/mL) but are reduced to 380 μg/mL, in compound **6**. This effect is even more notable in compound **7**, where a methyl group decreased the IC_50_ value from 325 μg/mL (compound **3**) to less than 50 μg/mL. Following our above discussion, a methyl group and a prenyl chain attached in a relative ortho position probably increases the lipophilic character of these compounds, and therefore enhances causing their accumulation in the fungal membrane. This increase not only compromises the membrane integrity but also contributes to oxidative stress through the generation of reactive oxygen species (ROS). This mechanism is well documented for phenolic compounds such as eugenol and its derivatives, where membrane disruption enhances redox activity, leading to increased intracellular ROS levels. It has also been shown that ROS fungal cell damage, which includes lipid peroxidation, harm to proteins and DNA, and membrane disruption, leads to fungal cell death [38,39,40,41]. Thus, the strong antifungal activity observed for compound **7** can be explained in terms of its lipophilic nature, which enhances membrane interaction and consequently induces lethal oxidative processes within fungal cells. In this context, hydration of the prenyl chain in compounds **8** and **9** led to a significant decrease in lipophilicity and, additionally, the double bond disappears. Without this double bond, the electron transport across the chain necessary to form ROS species is reduced. Therefore, hydrated compounds should exhibit a lower effect on membrane integrity and a decrease on ROS production. This suggests that the presence of the conjugated double bond in the prenyl chain plays a crucial role in the antifungal efficacy of these derivatives.

Although no commercial fungicide was included as a positive control, the literature reports allow a comparison with natural product standards. For example, thymol inhibits *B. cinerea* spore germination with IC_50_ ≈ 19.5 µg/mL [42]. In this context, compound **7** shows comparable antifungal activity (IC_50_ < 50 µg/mL), while maintaining low phytotoxicity on tomato (*Solanum lycopersicum* L.) seeds and roots. These results suggest that prenylated phenols could serve as promising scaffolds for developing natural-inspired antifungal agents with improved selectivity and safety

#### 2.2.2. Phytotoxic Activity

Compounds that showed activity against *B. cinerea* spore germination (**2**, **3**, **5**, and **7**) were also evaluated for their phytotoxic effects on tomato (*Solanum lycopersicum* L.) seeds and seedlings. Specifically, their impact on seed germination and root growth was assessed. The experimental conditions are detailed in Section 3.2.2, and the results obtained for compound **6** in the tomato (*Solanum lycopersicum* L.) seed germination assay are presented in Appendix A Appendix A.

Seed germination rate was measured in the presence of different concentrations of active compound, and the results are shown in Figure 2.

Germination of tomato (*Solanum lycopersicum* L.) seeds in presence of the highest tested concentration (200 μg/mL) of derivatives **2**, **3** and **5** is completely or almost completely inhibited. To assess if there is a time effect on inhibition, germination rates were determined from 24 h to 168 h, at 200 μg/mL concentration of each compound. The results are shown in Figure 3.

Results shown in Figure 2 and Figure 3 indicate that **2** and **3** are the most phytotoxic prenyl derivatives. By decreasing concentrations, the inhibition effect decreases and almost disappears for compounds **6** and **7**, but compounds **2** and **5** still exhibit important inhibitory germination activity at 50 μg/mL. After 168 h the phytotoxic effect of compounds **5**–**7** is completely lost, whereas compounds **2** and **3** remain active.

Root length serves as a critical indicator of plant health; thus, any compound impacting plant development is expected to affect root growth as well—shorter roots are indicative of stronger phytotoxic effects. Tomato (*Solanum lycopersicum* L.) root growth was assessed on the seventh day following the completion of germination and quantified by measuring root length using ImageJ software [42]. The results for compound **3** are presented in Appendix A Appendix A.

Root length was measured in presence of three different concentrations of each tested compounds and using ethanol as negative control. From these results, root growth percentages were calculated, and plots of percentage growth ± standard deviation were plotted as a function of concentration for each tested compound. The results are shown in Figure 4.

Results show that root growth inhibition by these compounds is a concentration-dependent process, and at the highest concentration (200 μg/mL) of compounds **2** and **3,** no root growth is observed. Additionally, for all tested concentrations, **2** and **3** are the most phytotoxic compounds in terms of root growth inhibition. This result is in line with the observed effect on seed germination, suggesting that these effects are related and inhibition of seed germination would negatively impact the root length.

To get a better understanding of phytotoxicity in terms of the chemical structure of tested compounds, results on inhibition of seed germination and root growth must be analyzed at the same time.

Interestingly, the addition of a methyl group to catechol derivatives (**7**) and resorcinol derivatives (**5**, **6**) decreases inhibition of germination seeds and root growth. In other words, methylation of aromatic ring in catechol or resorcinol derivatives reduces their phytotoxic activity. For example, in the presence of 200 μg/mL of these compounds, the level of germination rate of tomato (*Solanum lycopersicum* L.) seeds is completely recovered after 168 h. This effect is more important when the methyl group is located at *ortho* position, relative to the prenyl chain (**6** and **7**).

On the other hand, the introduction of a methyl group in the catechol structure seems to mitigate the phytotoxicity of these compounds on tomato (*Solanum lycopersicum* L.) seeds and, at the same time, increases their cytotoxicity on *B. cinerea* spores. This difference indicates clearly that their interaction with both biological systems follows completely different mechanisms.

Anyway, these results imply that both molecular structure and functional groups play a significant role in modulating the biological activity of these compounds. For example, increasing the lipophilicity of these molecules may enhance their interaction with fungus membrane permeability, and consequently an increase in antifungal activity is observed. On the other hand, this highest lipophilic character decreases their phytotoxicity, possibly due to altered solubility or the ability to interact with biological targets.

In summary, methylation of catechol and resorcinol derivatives enhances antifungal activity against *B. cinerea* of prenylated phenols, and, at the same time, decreases their phytotoxicity. In this context, compound **7** appears as a potential antifungal agent, with almost null phytotoxicity against seeds and roots of tomato.

These results highlight the importance of structural modifications in modulating the biological effects of compounds, with potential applications in plant growth regulation or crop protection.

## 3. Materials and Methods

### 3.1. Synthesis and Characterization

#### 3.1.1. General Procedures

Unless otherwise indicated, all purchased chemical reagents (Sigma-Aldrich, Merck, Darmstadt, Alemania and Fluka, Charlotte, NC, USA) were of the highest commercial purity and were used without prior purification. The ^1^H, ^13^C, ^13^C DEPT-135, sel. gs1D ^1^H NOESY, gs2D HSQC, and gs2D HMBC NMR spectra were recorded in solutions of CDCl_3_ or (CD_3_)_2_CO and are referenced to the residual peaks at δ = 7.26 ppm and δ = 77.0 ppm for CDCl_3_ and at δ = 2.05 ppm and δ = 29.9; δ = 206.7 ppm for (CD_3_)_2_CO for ^1^H and ^13^C, respectively. All analyses were performed on a Bruker NEO Avance 400 digital NMR spectrometer (Bruker, Rheinstetten, Germany), operating at 400.1 MHz for ^1^H and 100.6 MHz for ^13^C. The chemical shifts are expressed in δ ppm, and the coupling constants (*J*) in Hz. GC–MS was carried out using a SHIMADZU GCMS-QP2010 instrument (Tokyo, Japan). Silica gel 60 F_254_ plates were used for TLC. Spots on TLC plates were detected by heating after spraying with 10% H_2_SO_4_ in H_2_O. Microwave syntheses were carried out in a microwave reactor (Monowave 200, Anton Parr, Seiersberg, Austria) using sealed glass microwave containers with polyether ether ketone (PEEK) lids (capacity 10 or 30 mL).

#### 3.1.2. Microwave-Assisted Electrophilic Aromatic Substitution Reaction

The coupling of prenol or allyl chloride and substituted phenols was carried out in the presence of ZnCl_2_ as catalyst and ethyl acetate as solvents, under microwave radiation. In a typical reaction, prenol or allyl chloride was slowly added to a solution of phenol and ZnCl_2_ in ethyl acetate while stirring at room temperature. Once the mixture became homogeneous, it was transferred to a microwave reactor and heated to 60 °C for 30 min. After this period, the power was turned off, and the temperature decreased to 40 °C. Water (3 × 20 mL) was slowly added to the reaction mixture and then extracted with ethyl acetate (EtOAc, 3 × 30 mL). Subsequently, the organic layer was dried over anhydrous MgSO_4_ and concentrated by distillation under reduced pressure. The reaction mixture was purified by preparative flash chromatography using silica gel cartridges (Merck 0.032–0.063 mm) and a mobile phase consisting of a mixture of hexane and ethyl acetate with increasing polarity (from 100% Hex: 0% AcOEt → 0% Hex: 100% AcOEt).

*2-(3-methylbut-2-en-1-yl)benzene-1,4-diol* (**1**).

Compound **1** was obtained by a microwave-assisted coupling reaction from hydroquinone (2.0 g, 18.2 mmol), prenol (0.78 g, 9.1 mmol), and ZnCl_2_ (4.95 g, 36.3 mmol) in 10 mL of ethyl acetate. Compound **1** was obtained as a white solid (221.3 mg, 13.7% yield). The spectroscopic data as well as melting point obtained were consistent with those reported previously [16].

*4-(3-methylbut-2-en-1-yl)benzene-1,3-diol* (**2**).

Compound **2** was obtained by a microwave-assisted coupling reaction from resorcinol (1.0 g, 9.1 mmol), prenol (0.78 g, 9.1 mmol), and ZnCl_2_ (2.47 g, 18.2 mmol) in 10 mL of ethyl acetate. Compound **2** was obtained as an orange semi-solid (304.2 mg, 18.8% yield). The spectroscopic data as well as the melting point obtained for **2** were consistent with those reported previously [16].

*4-(3-methylbut-2-en-1-yl)benzene-1,2-diol* (**3**).

Compound **3** was obtained by a microwave-assisted coupling reaction from pyrocatechol (2.0 g, 18.2 mmol), prenol (1.56 g, 18.2 mmol), and ZnCl_2_ (4.95 g, 36.3 mmol) in 10 mL of ethyl acetate. Compound **3** was obtained as a white solid (143.4 mg, 4.4% yield). The spectroscopic data and the melting point obtained for **3** were consistent with those reported previously [16].

*2-methyl-5-(3-methylbut-2-en-1-yl)benzene-1,4-diol* (**4**).

Compound **4** was obtained through a microwave-assisted electrophilic aromatic substitution reaction using methylhydroquinone (1.0 g, 8.1 mmol), prenol (0.69 g, 8.1 mmol), and ZnCl_2_ (2.20 g, 16.1 mmol) in 10 mL of ethyl acetate. Compound **4** (73.1 mg, 2.4% yield) was obtained as a white semi-solid. ^1^H-RMN (CDCl_3_): 6.55 (s, 2H, H-3 y H-6); 5.11 (qt, *J* = 6.8 Hz y 1.4 Hz, 1H, H-2′); 4.68 (s, 1H, OH-4); 4.44 (s, 1H, OH-1); 3.37 (dt, *J* = 6.8 Hz y 1.0 Hz, 2H, H-2′); 2.19 (s, 3H, CH_3_-2); 1.80 (d, *J* = 1.2 Hz, 3H, H-4′); 1.72 (d, *J* = 1.3 Hz, 3H, H-5′). ^13^C-RMN (CDCl_3_): 148.0 (C-4); 147.7 (C-1); 133.6 (C-3′); 127.4 (C-5); 123.4 (C-2); 121.7 (C-2′); 113.4 (C-3); 113.2 (C-6); 26.1 (C-1′); 25.7 (C-4′); 18.0 (C-5′); 12.0 (CH_3_-2), Appendix A. EM (*m*/*z*, %): 191.11 (M^+^, 100.0), Appendix A.

*2-methyl-4-(3-methylbut-2-en-1-yl)benzene-1,3-diol* (**5**).

Compound **5** was obtained through a microwave-assisted electrophilic aromatic substitution reaction using 2-methylresorcinol (2.0 g, 16.1 mmol), prenol (1.39 g, 16.1 mmol), and ZnCl_2_ (4.39 g, 32.2 mmol) in 10 mL of ethyl acetate. Compound **5** (997.4 mg, 42.9% yield) was obtained as an orange semi-solid. ^1^H-RMN (CDCl_3_): 6.80 (dd, *J* = 8.8 Hz y 0.6 Hz, 1H, H-5); 6.34 (d, *J* = 8.4 Hz, 1H, H-6); 5.33 (s, 1H, OH-3); 5.30 (qt, *J* = 7.2 Hz y 1.5 Hz, 1H, H-2′); 3.30 (d, *J* = 7.2 Hz, 2H, H-1′); 2.14 (s, 3H, CH_3_-2); 1.80 (d, *J* = 0.4 Hz, 3H, H-4′); 1.78 (d, *J* = 1.2 Hz, 3H, H-5′). ^13^C-RMN (CDCl_3_): 153.4 (C-3); 153.2 (C-1); 135.0 (C-3′); 126.8 (C-5); 122.2 (C-2′); 118.4 (C-4); 110.9 (C-2); 107.0 (C-6); 30.0 (C-1′); 25.8 (C-4′); 17.8 (C-5′); 8.1 (CH_3_-2). EM (*m*/*z*, %): 191.11 (M^+^, 100.0).

*5-methyl-2-(3-methylbut-2-en-1-yl)benzene-1,3-diol* (**6**).

Compound **6** was obtained by a microwave-assisted coupling reaction from orcinol (1.0 g, 8.1 mmol), prenol (0.69 g, 8.1 mmol), and ZnCl_2_ (2.20 g, 16.12 mmol) in 10 mL of ethyl acetate. Compound **6** was obtained as an orange solid (182.5 mg, 11.8% yield). The spectroscopic and melting point data obtained for **6** were consistent with those reported previously [16].

*4-methyl-5-(3-methylbut-2-en-1yl)benzene-1,2-diol* (**7**).

Compound **7** was obtained through a microwave-assisted electrophilic aromatic substitution reaction using 4-methylcatechol (2.0 g, 16.1 mmol), prenol (1.39 g, 16.1 mmol), and ZnCl_2_ (4.39 g, 32.2 mmol) in 10 mL of ethyl acetate. Compound **7** (539.8 mg, 17.4% yield) was obtained as a white semi-solid. ^1^H-RMN (CDCl_3_): 6.67 (s, 2H, H-3 y H-6); 5.20 (qt, *J* = 7.14 Hz y 1.4 Hz, 1H, H-2′); 5.03 (s, 1H, OH-2); 5.00 (s, 1H, OH-1); 3.17 (d, *J* = 7.2 Hz, 2H, H-1′); 2.17 (s, 3H, CH_3_-4); 1.74 (d, *J* = 1.2 Hz, 3H, H-4′); 1.70 (s, 3H, H-5′). ^13^C-RMN (CDCl_3_): 141.1 (C-1); 141.1 (C-2); 132.6 (C-5); 132.4 (C-3′); 128.6 (C-3); 122.6 (C-2′); 117.2 (C-6); 115.8 (C-4); 31.3 (C-1′); 25.7 (C-4′); 18.7 (CH_3_-4); 17.8 (C-5′), Appendix A. EM (*m*/*z*, %): 191.11 (M^+^, 100.0), Appendix A.

#### 3.1.3. Hydration Reaction

Hydration reactions were performed by dissolving prenylated phenols in a mixture of 20 mL PEG-400 and 40 mL of distilled water and using *p*-toluenesulfonic acid as catalyst (PTSA). The reaction mixture was refluxed with constant stirring for 30 min at 100 °C. The completion of the reaction was monitored by TLC. The resulting mixture was washed with distilled H_2_O (3 × 20 mL). The organic phase was extracted using AcOEt (3 × 30 mL) and subsequently dried with anhydrous MgSO_4_, filtered, and evaporated. The reaction was purified by preparative flash chromatography using cartridges with silica gel (Merck 0.032–0.063 mm) and as a mobile phase, a mixture of hexane/AcOEt of increasing polarity (100% Hex: 0% AcOEt → 0% Hex: 100% AcOEt).

*4-(3-hydroxy-3-methylbutyl)-5-methylbenzene-1,2-diol* (**8**).

Compound **8** was obtained by hydration of 4-methyl-5-(3-methylbut-2-en-1yl)benzene-1,2-diol (0.54 g, 2.8 mmol) in the presence of PTSA (0.90 g) in a mixture of PEG-400 and water. Compound **8** (150.3 mg, 25.6% yield) was obtained as a white solid. Melting point 131–133 °C. ^1^H-RMN ((CD_3_)_2_CO): 7.46 (s, 2H, OH-2 y OH-1); 6.62 (s, 1H, H-3); 6.60 (s, 1H, H-6); 3.45 (s, 1H, H-3″); 2.54 (m, 2H, H-2′); 2.13 (s, 3H, CH_3_-5); 1.62 (m, 2H, H-1′); 1.24 (s, 6H, H-4′ y H-5′). ^13^C-RMN ((CD_3_)_2_CO): 144.4 (C-1); 144.2 (C-2); 133.9 (C-4); 128.0 (C-5); 118.7 (C-6); 117.5 (C-3); 70.0 (C-3′); 46.7 (C-1′); 30.3 (C-4′ y C-5′); 29.0 (C-2′); 19.2 (CH_3_-5), Appendix A. EM (*m*/*z*, %): 209.12 (M^+^, 100.0), Appendix A.

*4-(3-hydroxy-3-methylbutyl)benzene-1,2-diol* (**9**).

Compound **9** was obtained by hydration of 4-(3-methylbut-2-en-1-yl)benzene-1,2-diol (0.5 g, 2.8 mmol) in the presence of PTSA (1.5 g) in a mixture of PEG-400 and water. Compound **9** (297.8 mg, 54.1% yield) was obtained as a white solid. Melting point 118–119 °C. ^1^H-RMN ((CD_3_)_2_CO): 7.63 (s, 1H, OH-2); 7.60 (s, 1H, OH-1); 6.71 (d, *J* = 8.0 Hz, 1H, H-6); 6.69 (d, *J* = 2.0 Hz, 1H, H-3); 6.52 (dd, *J* = 8.0 Hz y 2.0 Hz, 1H, H-5); 3.31 (s, 1H, H-3″); 2.55 (m, 2H, H-2′); 1.68 (m, 2H, H-1′); 1.21 (s, 6H, H-4′ y H-5′). ^13^C-RMN ((CD_3_)_2_CO): 145.7 (C-2); 143.6 (C-1); 135.7 (C-4); 120.2 (C-5); 116.1 (C-3); 115.9 (C-6); 70.0 (C-3′); 47.1 (C-1′); 30.2 (C-2′); 29.8 (C-4′ y C-5′), Appendix A. EM (*m*/*z*, %): 195.10 (M^+^, 100.0), Appendix A.

### 3.2. Bioactivity Assays

The cytotoxicity of tested compounds on *B. cinerea* spores is a measure of their potential antifungal activity. Then, phytotoxic effects on tomato (*Solanum lycopersicum* L.) seeds and roots are evaluated for those exhibiting the highest activities on germination of *B. cinerea* spores. This information will allow us to decide if antifungal compounds can be used without affecting the plants that are intended to be protected from the fungus.

#### 3.2.1. Cytotoxicity on *B. cinerea* Spores

Preparation of culture media. The culture media for *B. cinerea* spores is prepared by dissolving, in water (500 mL), a mixture of Roswell Park Memorial Institute (RPMI) medium (5.210 g) and MOPS (17.25 g). The pH of the culture medium is adjusted to 7.0 using NaOH solution (10 N). The dissolution is then autoclaved for 40 min at 121 °C. Once cooled, 1000 μL of antibiotic stock (1%) is added. The RPMI-MOPS culture medium is stored and frozen until use.

Sample stocks of tested compounds are prepared at a concentration of 10 µg/µL in DMSO. Then, a volume of this dissolution (20 μL) is diluted with sterilized water (780 μL) to give a final concentration of 250 µg/mL in a 2.5% DMSO aqueous solution. The negative control was a 2.5% DMSO aqueous solution.

Preparation of developer solutions. MTT (thiazolyl blue tetrazolium bromide 98%, (Across Organics, Geel, Amberes, Belgium) (40.3 mg) and menadione (1.6 mg) are weighed into a falcon tube, and sterilized RPMI-MOPS solution (8 mL) are added. The resulting developer dissolution is MTT at 5 mg/mL and menadione at 1.16 mM.

Furthermore, an acidic isopropanol solution is prepared by mixing isopropanol (71.25 mL) with concentrated HCl (3.75 mL).

Cytotoxicity assay on *B. cinerea* spores. The antifungal activity of the synthesized compounds is evaluated using a modified spore growth and germination inhibition assay on spores *of B. cinerea* B-05 [42,43]. Spores of *B. cinerea* B-05 were obtained from the fungus collection of the Institute of Agricultural Sciences-CSIC, Madrid, Spain. Stock dissolutions of compounds are tested at final concentrations of 200, 100, and 50 µg/mL. The spore suspensions of *B. cinerea* are prepared in distilled water (1 × 10^7^ cells/mL) [42,44]. The samples and spore suspensions (4 replicates) are placed in 96-well plates and incubated for 24 h at 25 °C. After the incubation process, MTT/menadione developer solution (25 µL) is added, and the plates are incubated for additional 3 h. After this time, acidic isopropanol solution (200 µL) is added and the plates are incubated for 30 min. Finally, the absorbance is read at 490 nm in an ELISA reader [35,45]. The inhibition of *B. cinerea* spore germination for each compound is calculated as$$inhibition\%=\frac{Abs\left(negative\;control\right)-Abs(tested\;compound)}{Abs\left(negative\;control\right)}\times 100$$

Values of inhibition percentages calculated for each concentration are used to generate a dose–response graph, i.e., inhibition percentage as function of log (conc). From these plots the IC_50_ values (the effective dose to produce 50% inhibition) are obtained using a regression curve (Statgraphics Centurion XVI, v. 16.1.02, Statgraphics Technologies, Inc., The Plains, VI, USA). Thymol (Sigma-Aldrich) is used as positive control with an IC_50_ value of 19.54 µg/mL (22.94–15.74 95% Confidence Limits).

#### 3.2.2. Phytotoxic Activity on Tomato (*Solanum lycopersicum* L.) Seeds

The phytotoxic effects of compounds on tomato (*Solanum lycopersicum* L.) seed germination and root growth are assessed for compounds exhibiting the highest activities on germination of *B. cinerea* spores. These experiments are carried out according to a reported methodology [42] with tomato (*Solanum lycopersicum* L.) seeds. The seeds are placed in 12-well microplates (4 wells with 10 seeds for each test) and stock solutions of tested compounds are added to reach final concentrations equal to 200, 100, and 50 µg/mL in the well. In addition, a control of EtOH/H_2_O (20 µL/500 µL) is included as well. Germination is monitored for 7 days, during which the seeds are exposed to a controlled environment of light/darkness (16 h light/8 h darkness per cycle) and temperature (23.5 °C during the day and 21 °C at night). The germination rate is measured after 96 h–7 days, whereas the tomato (*Solanum lycopersicum* L.) root lengths are at the end of the experiment (7 days). Then, twenty-five tomato (*Solanum lycopersicum* L.) seedlings are randomly selected, and root length is measured using ImageJ software (v. 1.54p, NIH, Bethesda, MD, USA), (https://imagej.net/ij/) [42]. Juglone (Sigma-Aldrich) is used as positive control (100% inhibition of all parameters).

The percentage of seed germination percentage is calculated as (G − C^+^)/(C^+^) × 100%, where G and C^+^ are the number of germinated seeds in the treatment and control.

The percentage of root growth is calculated as (L − C^+^)/(C^+^) × 100%, where L and C^+^ are the root lengths measured for the treatment and control.

#### 3.2.3. Statistical Analysis

All data were expressed as the mean ± standard deviation (SD). Due to non-parametric data, a Kruskal–Wallis ANOVA was used with STATISTIC 7.0 program. Statistical significance was defined as *p* < 0.05.

## 4. Conclusions

In this study, a series of prenylated phenol derivatives have been synthesized and characterized. From these, **4**, **5**, and **7** are new compounds. Additionally, hydrated derivatives **8** and **9** were synthesized as well.

The antifungal activity of these derivatives was evaluated by measuring inhibition of spore germination of *B. cinerea*. Results indicate that the addition of a methyl group to the aromatic ring enhanced the activity of the compounds. This suggests that the introduction of the methyl group increases the lipophilicity of the molecule. Furthermore, the presence of the allyl chain was found to improve the compounds’ activity in inhibiting the germination of *B. cinerea* spores (B-05). On the other hand, hydration of the prenyl side chain showed no activity, suggesting that the presence of the double bond is important for antifungal activity. The double bond likely participates in redox processes with biological targets, leading to the production of reactive oxygen species (ROS) that contribute to the destruction of the cell membrane. This mechanism may disrupt cellular integrity, impairing vital processes such as membrane transport and enzyme function. Additionally, the generation of ROS can trigger oxidative stress, further enhancing the antimicrobial effect by damaging cellular components, including proteins, lipids, and nucleic acids

Finally, the results showed that the catechol derivatives exhibited the best activity, as indicated by their IC_50_ values. As previously mentioned, these derivatives likely create two distinct regions of marked polarity within the molecule, allowing the apolar region—represented by the prenyl chain—to interact more effectively with, or even penetrate, the fungal membrane, thereby causing the observed cellular damage.

The high antifungal activity shown by compound **7**, at low concentrations, suggests that this compound could be a promising candidate for further development as a potential sporicidal. These results could have significant implications for sustainable plant disease control and crop protection, where environmentally friendly and targeted approaches are essential.

On the other hand, compounds **2** and **3** exhibit important activities in seed germination and root growth, especially at a concentration of 200 μg/mL.

Further research is needed to optimize the synthesis and purification of these compounds, as well as to evaluate their biosafety and efficacy on a broader range of crops and pathogens. Future research should confirm this mechanism through direct ROS measurements and evaluate its efficacy under field conditions.

## Data Availability

Data are contained within the article and Appendix A.

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
