# Peer review of "Evaluation of Phytotoxic and Cytotoxic Effects of Prenylated Phenol Derivatives on Tomato Plants (*Solanum lycopersicum* L.) and *Botrytis cinerea* B-05 Spores"

_plants, 2025, doi:10.3390/plants14213277_

Round 1
Reviewer 1 Report
Comments and Suggestions for Authors
This manuscript reports the synthesis of a series of prenylated and hydrated phenol derivatives and the evaluation of their antifungal and phytotoxic activities, particularly against Botrytis cinerea and tomato seedlings. The topic is relevant to the search for safer and more sustainable alternatives to chemical fungicides. The experiments are well organized, and the combination of synthetic chemistry with biological assays gives the study a solid interdisciplinary character.
However, while the work presents promising findings, it still needs significant improvement before it can be accepted for publication.
The manuscript requires thorough English revision by a fluent or professional editor. Many sentences are awkward or overly long, and some terms are used inconsistently (e.g., “fungicides molecules”, “hydrated prenylphenols”). Improved phrasing will make the text more concise and readable.
The introduction should make clearer what is novel in this study compared with previous work by the same or related groups. Several earlier publications from the authors on eugenol and allylphenol derivatives are cited, but the distinct contribution of this paper (e.g., the specific role of methyl substitution or hydration effects) needs to be explicitly highlighted.
-
The synthesis section would benefit from reaction yields for all compounds, not just selected examples.
-
The biological assay description should specify how the concentrations were selected and whether controls for solvent toxicity were included.
-
Details about statistical analysis (number of replicates, exact test parameters) should be moved or expanded in the Methods section rather than repeated in figure captions.
Figures 3–5 are difficult to read; axes labels and concentration units should be enlarged and standardized.
-
A table summarizing IC₅₀ values alongside phytotoxic effects would help compare structure–activity relationships at a glance.
-
The supplementary material should be referenced more precisely in the main text (e.g., “Figure S6 shows…”).
-
The discussion occasionally repeats results instead of synthesizing them. A more focused interpretation comparing antifungal and phytotoxic effects structurally would be more compelling.
-
The explanation invoking ROS generation is interesting but speculative; supporting evidence or literature references specific to these phenolic derivatives would strengthen the argument.
-
The ecological or applied significance (for example, possible use in integrated pest management) could be better emphasized in the conclusion.
-
Some compound numbering and reference formatting are inconsistent.
-
Figure legends should be self-contained and define all abbreviations.
Author Response
Reviewer 1
The paper entitled “Evaluation of Phytotoxic and Cytotoxic Effects of Prenylated Phenol Derivatives on Tomato Plants and Botrytis cinerea B-05 Spores” describes a synthesis of prenylphenols and hydrated prenylphenols and the study of their effect on B. cinerea spore germination, as well as phytotoxic effect on tomato seed germination and root growth. The topic of this Ms seems to be original and new because the search of new natural product derivatives active against phytopathogenic fungus B.cinerea that causes gray mold disease is actual. Currently this pathogen is a limiting factor on agricultural production and to many chemical fungicides a resistant population of B.cinerea has appeared. The main question addressed by this research is the development of new natural-derived compounds, especially with prenylphenol unit, that will help to overcome the above mentioned problem. The authors have found that compound 4 exhibit good activity against B. cinerea B-05 spores being innocuous against tomato seed germination and root growth that all togerther makes it a potential sporicidal agent. The conclusions in this Ms are detailed, the necessary arguments are presented and addressed to the main question posed.
The paper could be recommended for publication after some revisions:
1) The similar structures at Fig. 1 should be combined using R substitution;
We have restructured Figure 1 according to this reviewer suggestion.
2) at Scheme 1 please add the numbering of all compounds;
We do not understand this reviewer’s request. In this scheme the general coupling reaction between 2-methylresorcinol and prenol is illustrated. Thus, the only compound that needs a number is the prenylated phenol, i.e. compound 5. The reactants are unique and so common that there is no need for a number.
3) why chemical fungicide was not used as a control in Table 1.
We agree with the Reviewer in this point. So, to solve this flaw, we have incorporated a new paragraph in Lines 206-212, in which we compare our results with that reported for thymol, i.e. IC₅₀ value of approximately 19.5 μg/mL for inhibition of Botrytis cinerea spore germination. This compound was selected because the measurements were performed under similar microplate assay conditions (Ref. 44). Comparison of IC50 values obtained for thymol and our most active compound (7, IC₅₀ < 50 μg/mL) is also discussed.
4) please explain why compound 4 is named as a perspective candidate in the Abstract while in Conclusion it is a compound 7.
We thank to the reviewer for pointing this out. The number used in the Abstract is wrong and therefore, we have now corrected it.
5) the melting point and optical rotation should be added for newly synthesized compounds.
We agree with the reviewer in the importance of these data. The melting points of all newly synthesized solid compounds have been added to the revised manuscript. However, for the semi-solid products, which are obtained as viscous oils, it was not possible to determine a melting point. Regarding the optical rotation, the synthesized derivatives are optically inactive and, therefore, optical rotation measurements are not applicable.
Reviewer 2 Report
Comments and Suggestions for Authors
The paper entitled “Evaluation of Phytotoxic and Cytotoxic Effects of Prenylated Phenol Derivatives on Tomato Plants and Botrytis cinerea B-05 Spores” describes a synthesis of prenylphenols and hydrated prenylphenols and the study of their effect on B. cinerea spore germination, as well as phytotoxic effect on tomato seed germination and root growth. The topic of this Ms seems to be original and new because the search of new natural product derivatives active against phytopathogenic fungus B.cinerea that causes gray mold disease is actual. Currently this pathogen is a limiting factor on agricultural production and to many chemical fungicides a resistant population of B.cinerea has appeared. The main question addressed by this research is the development of new natural-derived compounds, especially with prenylphenol unit, that will help to overcome the above mentioned problem. The authors have found that compound 4 exhibit good activity against B. cinerea B-05 spores being innocuous against tomato seed germination and root growth that all togerther makes it a potential sporicidal agent. The conclusions in this Ms are detailed, the necessary arguments are presented and addressed to the main question posed. The paper could be recommended for publication after some revisions: 1) The similar structures at Fig. 1 should be combined using R substitution; 2) at Scheme 1 please add the numbering of all compounds; 3) why chemical fungicide was not used as a control in Table 1. 4) please explain why compound 4 is named as a perspective candidate in the Abstract while in Conclusion it is a compound 7. 5) the melting point and optical rotation should be added for newly synthesized compounds.
Author Response
Reviewer 2
The manuscript describes the potential new approach to control the phytopathogenic fungus Botrytis cinerea that causes gray mold disease on numerous important crop plants. This new approach is based on natural product-inspired compounds instead of conventional chemical fungicides. This aspect is very important and valuable in the context of the growing resistance of phytopathogenic fungi to widely used fungicides.
The studied compounds are prenylphenols, and some hydrated prenylphenols. For the synthesis of the series of these compounds (including 3 new molecules), the use of microwave irradiation method was applied.
The obtained results are promising from the point of view of bioactivity, particularly as for inhibition of the germination of B. cinerea spores. Some interesting observations concerning the dependence of biological activity on the structure of the compounds were also made.
I consider this manuscript as interesting and valuable, however, I have several remarks.
- Both in the abstract and introduction, the clear statement of the aim is missing. The general idea is understandable; however, the main aim of the study should be clearly stated for the reader.
Thank you for this observation. Following this comment, we have rewritten the Abstract making explicit mention to the main aim of this study. The same idea has been carried out in the Introduction section (Lines 63-66).
- In the abstract, the Authors precisely listed the yields of biosynthesis of prenylphenols by microwave irradiation method, whereas no numeric data on their bioactivity is mentioned. Since the readers of the journal Plants are usually more biologists than only chemists, such data would be very appreciated.
Thank you for this valuable suggestion. As mentioned above, the whole abstract has been rewritten and the main data obtained for antifungal activity (IC50 values) have been added.
- The methods were well selected and described. However, in the test of B. cinerea spore inhibition, only negative control (with DMSO and water) were prepared. Usually in testing antifungal activity, also a positive control (e.g., with a commercial fungicide) is required. Since there was no positive control in this study, please at least discuss – on the basis on literature references – what is the efficiency of the tested compounds in comparison of the fungicides used in agriculture, or in any other laboratory experiments on inhibition of B. cinerea spores.
We agree with the reviewer in the importance of a positive control and following your advice we have added a discussion (Lines 206-212) in which our IC50 values are compared with that reported for thymol. The choice of this positive control was made because the measurement of its IC50 ≈ 19.5 μg/mL against Botrytis cinerea spore germination was carried out under similar microplate protocols (Ref. 44).
- Please correct the Latin names used in the text, sometimes they are not written in italics.
We have carefully revised the manuscript to ensure that all Latin names of species are written in italics according to scientific conventions. This includes Botrytis cinerea and Solanum lycopersicum.
Reviewer 3 Report
Comments and Suggestions for Authors
The manuscript describes the potential new approach to control the phytopathogenic fungus Botrytis cinerea that causes gray mold disease on numerous important crop plants. This new approach is based on natural product-inspired compounds instead of conventional chemical fungicides. This aspect is very important and valuable in the context of the growing resistance of phytopathogenic fungi to widely used fungicides.
The studied compounds are prenylphenols, and some hydrated prenylphenols. For the synthesis of the series of these compounds (including 3 new molecules), the use of microwave irradiation method was applied.
The obtained results are promising from the point of view of bioactivity, particularly as for inhibition of the germination of B. cinerea spores. Some interesting observations concerning the dependence of biological activity on the structure of the compounds were also made.
I consider this manuscript as interesting and valuable, however, I have several remarks.
- Both in the abstract and introduction, the clear statement of the aim is missing. The general idea is understandable, however, the main aim of the study should be clearly stated for the reader.
- In the abstract, the Authors precisely listed the yields of biosynthesis of prenylphenols by microwave irradiation method, whereas no numeric data on their bioactivity is mentioned. Since the readers of the journal Plants are usually more biologists than only chemists, such data would be very appreciated.
- The methods were well selected and described. However, in the test of B. cinerea spore inhibition, only negative control (with DMSO and water) were prepared. Usually in testing antifungal activity, also a positive control (e.g., with a commercial fungicide) is required. Since there was no positive control in this study, please at least discuss – on the basis on literature references – what is the efficiency of the tested compounds in comparison of the fungicides used in agriculture, or in any other laboratory experiments on inhibition of B. cinerea spores.
- Please correct the Latin names used in the text, sometimes they are not written in italics.
Author Response
Reviewer 3
This manuscript reports the synthesis of a series of prenylated and hydrated phenol derivatives and the evaluation of their antifungal and phytotoxic activities, particularly against Botrytis cinerea and tomato seedlings. The topic is relevant to the search for safer and more sustainable alternatives to chemical fungicides. The experiments are well organized, and the combination of synthetic chemistry with biological assays gives the study a solid interdisciplinary character.
However, while the work presents promising findings, it still needs significant improvement before it can be accepted for publication.
The manuscript requires thorough English revision by a fluent or professional editor. Many sentences are awkward or overly long, and some terms are used inconsistently (e.g., “fungicides molecules”, “hydrated prenylphenols”). Improved phrasing will make the text more concise and readable.
The introduction should make clearer what is novel in this study compared with previous work by the same or related groups. Several earlier publications from the authors on eugenol and allylphenol derivatives are cited, but the distinct contribution of this paper (e.g., the specific role of methyl substitution or hydration effects) needs to be explicitly highlighted.
- The synthesis section would benefit from reaction yields for all compounds, not just selected examples.
We are not entirely sure about this point, as the reaction yields for all synthesized compounds are already provided in the Experimental section.
- The biological assay description should specify how the concentrations were selected and whether controls for solvent toxicity were included.
The concentrations for the antifungal assay have been chosen following previous publications and adjusted to the antifungal range of our compounds.
The solvent controls are described in Materials and Methods (Lines 417-420 and 454-455)
- Details about statistical analysis (number of replicates, exact test parameters) should be moved or expanded in the Methods section rather than repeated in figure captions.
We completely agree with this point, and therefore a dedicated section on statistical analysis has been included in the Methods section (lines 468–471).
Additionally, replicates are described in Materials and methods Lines 431, 452 and 458-462. The figure captions of F4 have been modified and do not include any experimental parameter.
- Figures 3–5 are difficult to read; axes labels and concentration units should be enlarged and standardized.
Figures 3–5 have been improved as requested; axis labels and concentration units have been enlarged and standardized.
- A table summarizing IC₅₀ values alongside phytotoxic effects would help compare structure–activity relationships at a glance.
We appreciate your suggestion to include a table summarizing IC₅₀ values alongside phytotoxic effects. However, since the antifungal and phytotoxicity assays were conducted under different conditions, direct comparison in a single table would be scientifically inappropriate without extensive normalization. To maintain clarity and reflect the distinct nature of each assay, we present the data in separate tables and sections, ensuring accurate interpretation and methodological rigor.
- The supplementary material should be referenced more precisely in the main text (e.g., “Figure S6 shows…”).
The supplementary material has been referenced more precisely in the main text, as suggested by the reviewer.
- The discussion occasionally repeats results instead of synthesizing them. A more focused interpretation comparing antifungal and phytotoxic effects structurally would be more compelling.
The discussion has been significantly improved to provide a more focused interpretation, including a clearer structural comparison of antifungal and phytotoxic effects. We believe this revision addresses the reviewer’s concern.
- The explanation invoking ROS generation is interesting but speculative; supporting evidence or literature references specific to these phenolic derivatives would strengthen the argument.
A supporting paragraph with relevant literature references specific to these phenolic derivatives has been added to strengthen the argument regarding ROS generation. Lines 190-205.
- The ecological or applied significance (for example, possible use in integrated pest management) could be better emphasized in the conclusion.
The conclusion paragraph has been revised to better emphasize the ecological and applied significance of the findings, including their potential use in integrated pest management strategies. Lines 495-499.
- Some compound numbering and reference formatting are inconsistent.
This point was corrected.
- Figure legends should be self-contained and define all abbreviations.
The figure legends have been revised to be self-contained and now include definitions for all abbreviations.
Round 2
Reviewer 1 Report
Comments and Suggestions for Authors
I recommend to be accepted in current form.